# Endoluminal Procedures and Devices for Esophageal Tract Investigation: A Critical Review

**DOI:** 10.3390/s23218858

**Published:** 2023-10-31

**Authors:** Giorgia Spreafico, Marcello Chiurazzi, Davide Bagnoli, Sergio Emiliani, Nicola de Bortoli, Gastone Ciuti

**Affiliations:** 1The BioRobotics Institute, Scuola Superiore Sant’Anna, 56127 Pisa, Italy; marcello.chiurazzi@santannapisa.it (M.C.); gastone.ciuti@santannapisa.it (G.C.); 2Department of Excellence in Robotics and AI, Scuola Superiore Sant’Anna, 56127 Pisa, Italy; 3Medica SpA, 40131 Bologna, Italy; davide.bagnoli@medica-spa.com (D.B.); sergio.emiliani@medica-spa.com (S.E.); 4Gastrointestinal Unit, Department of Translational Sciences and New Technologies in Medicine and Surgery, University of Pisa, 56124 Pisa, Italy; nicola.debortoli@unipi.it

**Keywords:** diagnosis, endoluminal devices, esophageal diseases, upper gastrointestinal tract investigation, esophageal diagnostic techniques

## Abstract

Diseases of the esophageal tract represent a heterogeneous class of pathological conditions for which diagnostic paradigms continue to emerge. In the last few decades, innovative diagnostic devices have been developed, and several attempts have been made to advance and standardize diagnostic algorithms to be compliant with medical procedures. To the best of our knowledge, a comprehensive review of the procedures and available technologies to investigate the esophageal tract was missing in the literature. Therefore, the proposed review aims to provide a comprehensive analysis of available endoluminal technologies and procedures to investigate esophagus health conditions. The proposed systematic review was performed using PubMed, Scopus, and Web of Science databases. Studies have been divided into categories based on the type of evaluation and measurement that the investigated technology provides. In detail, three main categories have been identified, i.e., endoluminal technologies for the (i) morphological, (ii) bio-mechanical, and (iii) electro-chemical evaluation of the esophagus.

## 1. Introduction

Clinical pathologies related to the esophageal tract are commonly seen in clinical practice. When considering esophageal conditions, there are many families of diseases involving an advanced interplay of factors; for instance, a few less severe pathological disorders, e.g., defective peristaltic behavior, may lead to frequent refluxate events and esophagus acidification, becoming the precursor of more serious conditions.

Although pathologies of the esophagus are heterogeneous and often correlated—as each of them contributes to the loss of a complex physiological balance—for the sake of simplicity, we may distinguish between (i) neoplastic and pre-neoplastic conditions, (ii) esophagitis, (iii) motility disorders, and (iv) gastro-esophageal reflux disease.

Among pre-neoplastic conditions, it is worth mentioning the Barrett’s esophagus (BE), a metaplastic change in the distal esophagus, whereby the normal squamous epithelium is replaced by columnar epithelium that has both gastric and intestinal features [1]; this condition develops because of acidification of the esophagus lumen, and it predisposes to the development of adenocarcinoma. Estimates of the prevalence of BE in the general population widely vary depending upon the population studied and the criteria used to establish the diagnosis, but it has been suggested that it may be under-estimated since BE remains undiagnosed in many asymptomatic cases [2]. Guidelines for diagnosis and management of BE are provided by the American College of Gastroenterology (ACG) with the intent of proposing recommendations to follow for screening and surveillance of patients with known BE [1].

For what concerns esophagitis, defined as irritation and inflammation of the esophageal inner walls, the most severe one is eosinophilic esophagitis (EoE), which is immune-allergic and has multifactorial etiology [3]. EoE is a quite rare disease, with a prevalence of ~42 cases out of 100,000 adults [4], but it leads to painful and difficult swallowing if it remains untreated. The most challenging task in this field is to characterize atypical clinical presentations of EoE that currently do not fulfill diagnostic criteria and, therefore, remain undiagnosed [5].

In the motility disorders domain, in which diseases are often chronic and dramatically affect patients’ quality of life, the best-defined motor esophageal disorder is achalasia, i.e., a disease characterized by defective esophageal body peristalsis and uncoordinated and uncomplete lower esophageal sphincter (LES) relaxation [6]. Achalasia results from the denervation of the intrinsic nervous system of the esophagus and it is considered a rare disease, with an annual incidence estimated at 1 case out of 100,000 adults [7].

Finally, gastroesophageal reflux disease (GERD)—a chronic condition defined in the Montreal Consensus as a condition that develops when the reflux of stomach contents comes into the esophagus, which causes troublesome symptoms and/or complications [8]—is the most prevalent gastrointestinal disorder, with an estimated worldwide prevalence in the range of 8–20% involving both genders [9]. Although many attempts have been made to quantitatively calculate the incidence and prevalence of GERD [10,11,12], such evaluations still remain uncertain due to difficult interpretations of symptoms.

Esophageal symptoms often overlap with different esophageal disorders, posing a challenge for patients’ healthcare management. However, the diagnosis of esophageal disorders has evolved, and several different techniques are now available. To establish the best treatment strategy, a combination of different diagnostic procedures is necessary to first provide evidence of a disease and then identify its causes.

This review does not intend to suggest a diagnostic procedure to follow but aims to inform readers with an overview of the available examination options, with their associated main features and devices; this work has been designed by bioengineers together with a clinician—N.d.B., author of the study—for medical appropriateness.

## 2. Materials and Methods

This systematic review follows the methodological guidelines of the PRISMA statement [13]; in detail, the PRISMA 2020 flow diagram is used for new systematic reviews.

The systematic literature research was performed using PubMed, Scopus, and Web of Science databases, and the following search queries were used: “(esophageal OR gastroesophageal) AND (function OR diseases) AND (testing OR investigation OR diagnosis) AND (device OR catheter OR system OR technology) NOT (treatment)”. Although the time interval included was 2013–today, previous works have been considered as “other sources” if relevant. The PRISMA diagram, outlining the literature review process, is presented in Figure 1.

### 2.1. Inclusion Criteria

Randomized controlled trials, clinical trials, observational studies, pilot studies, and retrospective studies that used endoluminal technologies to evaluate the esophagus health condition were included in the review. Only papers written in English have been accepted. Other inclusion criteria were: (i) only peer-reviewed papers were considered, (ii) only the most recent update of the same work was included, and (iii) only papers dealing with endoluminal technologies used for diagnosis inside the esophagus were accepted.

Eligibility criteria and study selection were carried out by one reviewer (G.S.) and supervised for decision checking by other reviewers (M.C. and G.C.). Disagreements were solved by discussion. Decisions were recorded using shared Mendeley (London, UK) libraries organized into folders and sub-folders containing included and excluded studies.

### 2.2. Strategies for Data Synthesis

Studies were divided into categories based on the type of evaluation and measurements that the investigated technology provides. In detail, three main categories were identified: (i) endoluminal technologies for morphological evaluations of the esophagus, (ii) endoluminal technologies for bio-mechanical evaluations of the esophagus; and (iii) endoluminal technologies for electro-chemical evaluations of the esophagus. Data were collected reporting the following information for each included study: (i) field of application of the presented technology, (ii) technology readiness level (TRL), (iii) design and working principle, (iv) pros and cons, and (v) future developments.

In Figure 2, the reader can find an overview of the main examined technologies; all of them were further detailed in the next sections.

## 3. Endoluminal Technologies for Morphological Evaluation of Esophagus

Examinations through visualization of the esophagus endoluminal morphology are the first-line techniques in the diagnosis of esophageal diseases. In detail, imaging techniques are indicated to assess the inspection of esophageal mucosa, to provide information about structural abnormalities, and to identify pathologies, such as BE, esophageal adenocarcinoma, and EoE. Several imaging modalities are available in the clinical practice, including—but not limited to—conventional white light endoscopy, narrow-band imaging, multispectral imaging, endomicroscopy, and other techniques described in detail in the next sub-sections. Each modality is characterized by different diagnostic performances (e.g., sensitivity, specificity, and accuracy) and comes with advantages and shortcomings in terms of availability, accuracy, tolerability, and costs. Direct visualization of anatomy and pathology of the esophagus lumen is often mandatory in the evaluation of the esophagus health condition, with the great advantage of giving evidence of chronic effects (not limited to the time window of observation). The following sub-sections provide an overview of the available modalities for the morphological evaluation of the esophageal tract, ranging from the most conventional ones to the newest—promising but with low TRL—techniques.

### 3.1. Conventional White Light Endoscopy (WLE)

Almost each procedure, focused on the management of esophageal symptoms, begins with an upper endoscopy to identify treatable etiologies and to rule out malignancies [17].

Conventional flexible endoscopes with small diameters (<10 mm) and large bending angles (up to 210 degrees) [18] are used to provide a real-time optical diagnosis by exploiting charge-coupled devices (CCDs) for image acquisition. Tissue features are represented in three colour bands, i.e., red (R: 620 ± 40 nm), green (G: 540 ± 40 nm), and blue (B: 470 ± 40 nm) for an RGB vision that replicates the spectral sensitivity of humans. Standard definition endoscopes are equipped with CCD chips that produce an image signal of 100–400,000 pixels, whereas high-definition endoscopes can achieve a resolution of more than 1 million pixels [18]. An external unit is responsible for recording, processing, and managing acquired images.

The procedure is conventionally performed by placing the endoscope trans-orally, and it often induces gagging reflexes in the patient, which may invalidate the diagnosis other than bringing to low tolerability of the procedure itself. In this regard, Mori et al. [19] compared the performances and the frequency of gagging reflexes with three different protocols: (i) oral endoscopy using a conventional endoscope (GIF XQ240 Olympus Corp., Tokyo, Japan) with a tip diameter of 9 mm, (ii) oral endoscopy using an ultra-thin endoscope (EG530N, Fujinon, Fuji Photo Film Co., Ltd., Tokyo, Japan) with a tip diameter of 5.9 mm, and (iii) trans-nasal endoscopy using the same ultra-thin endoscope mentioned above. Both the used gastroscopes are shown, respectively, in Figure 3(aI,II). A cohort of 1580 patients was enrolled in this study, and the diagnosis was made using endoscopic images collected by three senior endoscopists with more than 10 years of experience. The authors concluded that trans-nasal endoscopy had an equivalent performance in the diagnosis of reflux esophagitis and BE compared with trans-oral endoscopies. On the other hand, trans-nasal endoscopy was associated with better tolerability and less gagging effects, suggesting that its use may be beneficial in clinical practice.

Towards patient’s acceptance improvement, an attractive alternative to conventional endoscopy that may lead to more comfortable procedures is capsule-based endoscopy. In endoscopic capsule design, a good trade-off must be found between high image quality and other features such as size, power consumption, simple control interface, and frame rate. While conventional endoscopes can rely on external light sources and tethered cameras, the capsule image acquisition hardware is fully integrated inside the device and consists of an image sensor, illumination, lenses, electronics, and battery [14,20]. Capsule-based endoscopes may be: (i) wireless and locomoted by esophagogastric motility, e.g., PillCam^®^ UGI capsule (Medtronic Inc., Minneapolis, MN, USA) shown in Figure 3(aIII), (ii) wireless and magnetically actuated from outside, such as the MiroCam-Navi system (Intromedic Co., Ltd., Seoul, Republic of Korea) shown in Figure 3(aIV), and (iii) tethered capsule-like endoscopes (still at a research level, with few commercial exceptions such as the Bravo™ calibration-free Reflux Testing System of Medtronic Inc., Minneapolis, MN, USA) [21,22]. Although lower image resolution and frame acquisition rate are associated with non-tethered devices, many studies demonstrated that magnetically assisted capsule endoscopy (MACE) is safe, well tolerated by patients, and accurate in the diagnosis of the upper gastrointestinal (GI) tract diseases [22,23,24,25]. The main limitation of wireless MACE is associated with the rapid transit of the capsule throughout the esophagus, which may lead to partial or incomplete visualization of the lumen. To improve capsule transit time, two options are available: (i) the previously mentioned tethered version of capsule-like endoscopes, which may also be used trans-nasally for a demonstrated better tolerability [26], and (ii) the use of a removable string attached to the capsule, used to control the movement of the capsule up and down the esophagus [27].

Improvements in the resolution of imaging during endoscopic procedures have resulted in a significant increase of polyp detection rate. On the other hand, with conventional WLE, in-vivo discrimination between neoplastic and non-neoplastic polyps may result difficult, bringing to unnecessary biopsies and polypectomies as precautionary measures, even if most of the polyps are non-neoplastic [28]; this aspect highlights the necessity for advanced imaging techniques—discussed in the next sections—able to improve visualization of lesions and identification of polyps’ histology.

### 3.2. Narrow-Band Imaging (NBI)

Narrow-band imaging (NBI) is an optical technology that uses only wavelengths absorbed by hemoglobin (i.e., blue and green wavelengths) for maximizing contrast and enhancing the detail of certain aspects, i.e., vascularization, of the surface of the mucosa. Usually, NBI can be achieved either by using customized LEDs with transmitted peaks matching the desired wavelength, or by incorporating an electronically activated filter onto a conventional endoscope; switching from the WLE imaging mode to NBI is easily achieved by a button on the handle of the endoscope.

In [29], Gounella et al. designed and fabricated two optical filters to provide desired peaks at 415 nm and 540 nm using seven layers of thin-films of ${\mathrm{SiO}}_{2}$ and ${\mathrm{TiO}}_{2}$ obtained using sputtering technique. Fabry–Perot etalons were used, which were designed to have a resonance cavity separated by two parallel identical mirrors with specific transmittance to create resonance at a given wavelength. Such a wavelength is cavity-thickness dependent, as shown in Equation (1):l = 2·q·n·d·cos(α),(1)
where l is wavelength, q is the interference order, n is the refractive index of the medium inside the cavity, d is the distance between the mirrors in meters, and α is the angle between the incident light rays and the upper mirror. Materials and fabrication parameters were selected following this mathematical model to achieve the desired peak wavelengths.

Compared to WLE, the probability of missing a lesion with the NBI technique is reduced thanks to less blurred images of capillaries, which are visualized in greater detail. Moreover, Singh et al. [28] highlighted the clinical value of NBI in characterizing suspicious mucosal areas and predicting the polyps’ histology.

Moreover, NBI has also found use in the identification of BE. Singh et al. [30] conducted a feasibility study using the 190 series Exera III NBI system (Olympus Corp., Tokyo, Japan) with dual-focus magnification (DF) capabilities (Figure 3b), a technique that allows switching from normal-focus mode to near-focus mode with a single button. The author demonstrated that detection of BE was feasible with high sensitivity, and biopsies could be avoided in 86% of the areas imaged with the NBI-DF combined system.

### 3.3. Multispectral Imaging (MSI)

Multispectral imaging (MSI) is a technique that represents an evolution of conventional WLE since it allows for the extraction of additional information beyond the visible light range (i.e., RGB), such as infrared and ultraviolet, by using a larger number of spectral bands. Such a technique is successfully applied for the detection and analysis of easily reachable carcinomas, ex-vivo samples of hollow organ mucosal carcinomas, and histological samples. Nevertheless, a commercial endoscopic tool able to perform in-vivo MSI is still not available since it would require significative miniaturization of optics to be integrated as a “chip-on-tip” camera at the distal end of an endoscope.

In this regard, Waterhouse et al. [31] performed a first-in-human pilot study using a commercial fiberscope to relay imaging data from the upper GI tract to a snapshot MSI camera capable of collecting data from nine spectral bands (Figure 3c). The images, collected through MSI system on 20 patients who underwent the trial were examined through several learning-based methods for data classification. In this preliminary study, the authors were able to provide evidence that MSI has the potential to improve the detection of neoplasia during the surveillance of BE.

The MSI technique responds to the unmet clinical need for optical methods with improved diagnostic yields and lower cost per procedure, but it is not the only solution to predict polyps’ histology, as other promising methods are discussed in the next sections.

### 3.4. Autofluorescence Imaging (AFI)

The principle of Autofluorescence imaging (AFI) is based on the detection of natural tissue fluorescence emitted by specific endogenous molecules called fluorophores (i.e., collagen, nicotinamide, adenine dinucleotide, flavin, and porphyrins). Exciting such molecules with short-wavelength light leads, indeed, to the emission of fluorescent light of longer wavelength [32]. The overall fluorescence emission differs among normal, inflamed, and neoplastic tissue due to corresponding differences in fluorophore concentration, metabolic state, and/or spatial distribution. As such, autofluorescence measurement can be used as a useful marker to enable tissue differentiation in real-time during endoscopy.

Recently, AFI systems have become available as an integral part of trimodal imaging video endoscopes that combine WLE, NBI, and AFI in a single device. The endoscopist can alternate freely between the three modalities at any time using a switch on the scope. Such endoscopes are provided with two separate monochromatic CCD cameras located at the tip of the scope: one is used for WLI and NBI, whereas the other is dedicated to AFI. When the AFI mode is selected, tissue is illuminated by short-wavelength light (blue and green, sequentially), and AFI CCD camera receives back tissue autofluorescence and reflected green light (blue light is eliminated through a filter). Finally, a video processor integrates the sequentially captured images of autofluorescence and green reflectance and creates real-time images in which normal mucosa typically appears green, and dysplastic tissue appears dark purple (Figure 3d) [33]; an example of a commercially-available device is GIF-FQ260Z, produced by Olympus Corp. (Tokyo, Japan).

Xi Luo et al. [34] presented the results of a first prospective observational trial performed on 127 patients affected by reflux symptoms, to investigate whether AFI could be used to distinguish non-erosive reflux disease from functional heartburn. All the subjects were investigated with WLE-AFI combined endoscopy, and the ones suspected with non-erosive reflux disease also underwent multichannel intraluminal impedance and pH (MII-pH) monitoring (the gold standard procedure for evaluating non-erosive reflux disease), which will be further discussed in Section 5. As a result, the authors concluded that AFI may be a powerful tool to distinguish functional heartburn and non-erosive reflux disease, since it reached an accuracy of 90.5% compared to the WLI state-of-the-art procedure.

On the other hand, Wong Kee Song et al. [33] claimed that the AFI technique may be a precious tool when used in combination with other diagnostic modalities, but it still lacks specificity, and it is not ready to be used as a standalone procedure. As such, other techniques are also being investigated to improve specificity in the recognition of impaired endoluminal tissue.

### 3.5. Surface-Enhanced Raman Spectroscopy (SERS)

Raman spectroscopy relies upon the inelastic scattering of photons exchanging energy via molecular vibrations after irradiation with a light laser and provides details of the chemical composition and molecular structures in cells and tissues. This technique can be applied to endoluminal investigations as it has the potential to probe compositional differences between tissues during routine endoscopy. Intrinsic Raman endoscopy is currently being clinically explored to rule out malignancies in the upper GI tract.

Garai et al. [35] presented the design and in-vivo demonstration of an entirely new miniaturized and non-contact device able to perform Raman endoscopy as an accessory to clinical endoscopes (Figure 3e).

Nevertheless, this technique still faces many inherent limitations, such as the required acquisition times and, most importantly, the weakness of Raman signal, which is due not only to the small fraction of photons (approximately 1 in 106–108) undergoing Raman scattering but also to the strong fluorescence background in biological samples [36]. To overcome these limitations, Surface-Enhanced Raman Spectroscopy (SERS) has been recently investigated. In detail, surface-enhanced resonance Raman scattering nanoparticles (SERRS-NPs) are used to amplify Raman signal by many orders of magnitude (up to 1014) and enhance visualization of functional information about the investigated tissue.

Harmsen et al. [37] used the clinically validated dual-modal Raman endoscopy system developed by Garai et al. in [35] to perform a pre-clinical study and demonstrate that a single dose of a high-sensitivity SERRS-NP is sufficient to reliably detect precancerous GI lesions in animal models that closely mimic disease’s development in humans. The authors concluded that since the composition of SERRS-NPs was biocompatible, and the dual-modal Raman endoscopy system adopted was already clinically-validated in humans, contrast-enhanced Raman endoscopy could become in future an adjunct tool to WLE. Clinical translation would be of great interest since SERS only relies on an objective diagnostic parameter of disease, thus it is free from interobserver interpretation disagreement.

### 3.6. Endomicroscopy

A relatively new promising tool enabling the acquisition of real-time histology-like images from inside hollow organs (i.e., esophagus and colon) is endomicroscopy. Different techniques can be used for the following purposes:i.Optical coherence tomography (OCT) is a cross-sectional imaging technique with 10 µm resolution, in which the necessary optics are located and cantered inside a transparent balloon-based probe (with about 60 mm of length and 25 mm of diameter). The OCT laser beam is helically scanned over the balloon and maps the internal walls of the esophageal lumen (Figure 3(fI)) [38,39,40]. The OCT technique allows to perform high resolution imaging of internal microstructures by measuring the echo time delay and magnitude of backscattered light. Deeper in detail, a coupler is used to split and direct the light from a low-coherence source towards different arms of an interferometer: the reference arm, in which the light is backscattered by a reference mirror, and the sample arm, in which the sample to analyse is placed. The returning light from both arms is recombined at the coupler to generate an interference pattern that depends on the microstructures of the sample: light will be backscattered when it encounters an interface between materials of different refractive index. Suter et al. [41] performed a feasibility pilot study to demonstrate that in-vivo OCT-based volumetric laser endomicroscopy (VLE) is a valuable tool to guide biopsy site selection. Indeed, when addressing BE and neoplasia, endoscopic biopsy specimens are usually taken at random locations during endoscopy to perform histopathologic analysis of tissues; the authors demonstrated that VLE-guided biopsy of the esophagus is safe and reduces sampling errors, leading to better diagnostic performances and optimal patient management. Interpretation of VLE images, however, is time-consuming and complex, and, to deal with this difficulty, novel algorithms for computer-aided multi-frame analysis have been developed [42].ii.Confocal laser endomicroscopy (CLE) is based on tissue illumination with a low-power laser focused on a selected depth in the tissue of interest and collection of the reflected light refocused onto the detection system by the same lens. This modality allows to obtain high magnification and resolution images of the mucosal layer of the GI tract [43]. Currently, there are two types of CLE: (i) endoscope-based (eCLE) and (ii) probe-based CLE (pCLE) techniques (Figure 3(fII)). The first one has a laser probe attached to the distal tip of the endoscope, whereas, in the latter system, the probe is placed through the biopsy channel of an endoscope [44]. Li et al. [44] compared eCLE and pCLE in their diagnostic yield in different segments of the GI tract, concluding that, for the examination of the stomach and colon, pCLE is more flexible than eCLE with a shorter procedure time, whereas eCLE provides better image quality than pCLE for the observation of the esophageal tract.

It has been suggested that endomicroscopy can be considered a powerful tool to perform a diagnosis of BE without resorting to biopsy [45]. Kollar et al. compared the diagnostic accuracy of pCLE with standard biopsies in patients with visible esophageal or gastric lesions [45]. Making a comparison with the diagnosis determined from the resection specimen of 74 lesions, the authors obtained an overall diagnostic accuracy, with conventional biopsies, of 85% compared to the accuracy obtained with pCLE of 89%, suggesting that pCLE could be recognized as a standard method to provide an accurate histopathological diagnosis.

Furthermore, new opportunities of screening for internal diseases were unveiled by recent studies on painless procedures to acquire three-dimensional, microstructural images of the upper GI tract using swallowable, tethered capsule-based endomicroscopy systems (Figure 3(fIII)) [46,47,48,49,50,51,52].

**Figure 3 sensors-23-08858-f003:**
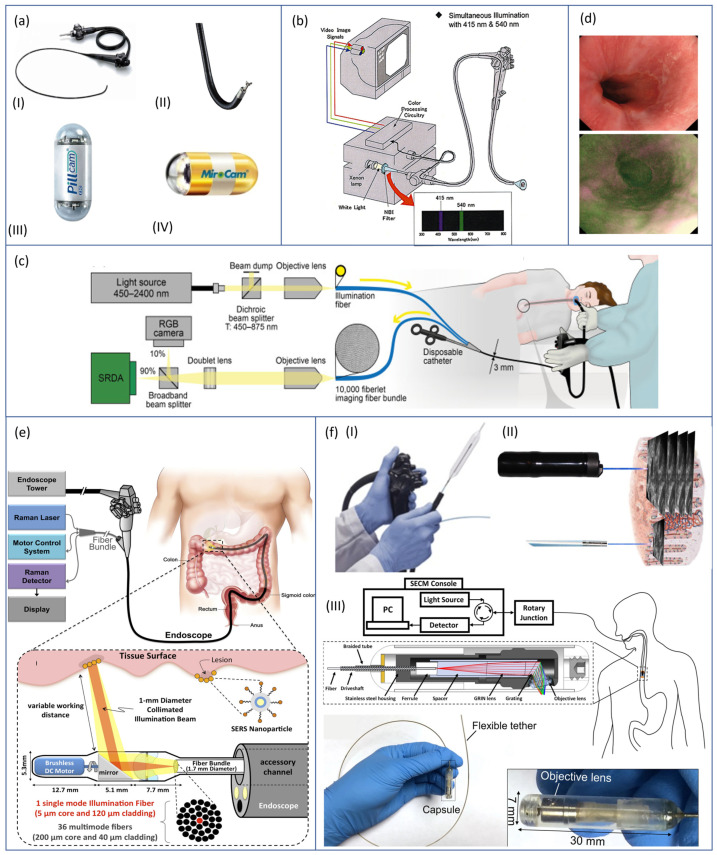
From the top left corner: (**a**) conventional White Light Endoscopic (WLE) devices, both tethered, (**I**) GIF XQ240 (Olympus Corp., Tokyo, Japan) (Courtesy of Olympus Corp.) and (**II**) EG530N (Fujinon, Fuji Photo Film Co., Ltd., Tokyo, Japan) (Courtesy of Fuji Photo Film Co., Ltd.), and untethered, (**III**) PillCam^®^ UGI capsule (Medtronic Inc., Minneapolis, MN, USA) [14] and (**IV**) MiroCam-Navi system (IntroMedic Co., Ltd., Seoul, Republic of Korea) [14]; (**b**) Exera III NBI system (Olympus Corp., Tokyo, Japan) with dual-focus magnification (DF) capability [53]; (**c**) conventional fiberscope provided with an external MSI system for in-vivo trial by Waterhouse et al. [31]; (**d**) comparison of normal and neoplastic tissue images captured with an AFI system [34]; (**e**) miniaturized and non-contact device able to perform Raman endoscopy, designed as an endoscopic add-on by Garai et al. [35]; and (**f**) endomicroscopy systems based, respectively, on: (**I**) OCT [50], (**II**) CLE [51], and (**III**) capsuled-based systems [52].

## 4. Endoluminal Technologies for Bio-Mechanical Evaluation of Esophagus

Although imaging techniques offer a great understanding of the health condition of the esophagus, such modalities lack in identifying non-visible abnormalities. To address this clinical need, endoluminal solutions able to evaluate bio-mechanical properties of the esophagus lumen are also available and presented in this section (Section 4). These technologies are used to reach a more complete understanding of esophageal motor functions, thus providing evidence of physiological or pathological contractility behaviours. The bio-mechanical evaluation of the esophageal tract is a functional study required when esophageal symptoms are not associated with visible morphological abnormalities—such as tightening or narrowing (e.g., strictures) of the esophagus lumen—and when other pathologies assessable with imaging procedures—such as EoE and esophageal adenocarcinoma—have been excluded by previous evaluations.

The following sub-sections report the available endoluminal technologies found in literature to evaluate esophageal bio-mechanical properties.

### 4.1. High-Resolution Manometry (HRM)

HRM is a diagnostic procedure performed to retrieve dynamic measurements of esophageal intraluminal pressure and characterize peristaltic pressure waves. Such treatment is currently used to distinguish physiological and pathological contractility behaviours.

The process of deglutition involves voluntary and reflexive activities of muscles and nerves. In order to successfully design a device able to accurately monitor such phenomenon, it is of great importance to deeply understand its kinematic. Many kinematic models of esophageal bio-mechanical behaviours can be found in the literature, mostly assuming that the esophageal tract is an axial symmetrical tube with finite length. Misra and Pandey in [54] proposed a simplified model that represents the peristaltic wave as a progressive sinusoidal wave, as described in Equation (2):
(2)$$h\left(z,t\right)\;=\;a\;-\;0.5\;\Phi \;\left[1\;+\;\frac{2\pi }{\lambda }\cos\left(z\;-\;\mathrm{ct}\right)\right]$$
where h is the radial displacement of the wave from the axis of esophagus, a is the radius of the esophagus, Φ is the amplitude of the wave, λ the wavelength, z is the axial distance along esophagus, c is the wave velocity, and t represents time.

The standard equipment for a HRM procedure consists of: (i) a manometric catheter with 36 pressure sensors and a 10 mm pitch (accordingly to the Chicago classification [55]), (ii) signal acquisition and processing units, and (iii) a graphical user interface (GUI) evaluating, in real-time, the sensor data for providing a spatial–temporal qualitative plot to the user (Figure 4a).

The procedure begins by placing the catheter trans-nasally into the patient until the tip reaches and passes both the upper esophageal sphincter (UES) and the lower esophageal sphincter (LES). Standard evaluation of esophageal motility was performed using 10 swallows of 5 cc of water in the supine position [56].

HRM is the evolution of a similar but less accurate procedure known as low-resolution manometry (LRM), which performs the same measurements with a lower number of sensors (between 5 and 8 sensors).

Two main categories of HRM systems exist, i.e., water-perfused catheters and solid-state ones, and they mainly differ in the pressure sensing methodology.

Water-perfused manometric catheters consist of an outer frame containing multiple channels. Each channel has an opening at a different point along the catheter lumen: an external perfusion pump drives distilled water into the channels, and pressures are transmitted back along the column of water from each opening point to the external transducer. Water-perfused catheters suffer from a low refresh rate, which is relevant when high dynamic phenomena must be monitored. Another drawback associated with water-perfused catheters relies on the time-consuming set up and the complexity of the system.

In solid-state manometry, pressure transducers are located inside the catheter itself, thus providing an improved dynamic response (update rate of 100 Hz in solid-state manometry vs. 20 Hz in water-perfused one) [57].

While water-perfused catheters provide only unidirectional sensors (measuring pressure at the point of the side hole), solid-state catheter sensors usually perform, circumferentially, integrated measurements. In this regard, it has recently been suggested that there may be significant differences between measurements provided by unidirectional and circumferential sensors. In the esophageal tract, there is evidence of radial symmetry in pressure, and this assumption is no longer valid in correspondence of radially asymmetric structures (e.g., UES and LES): for this reason, it seems reasonable that a circumferential system records more accurate data [56]. Hernandez et al. [58] investigated within-subjects differences between unidirectional and circumferential pressure measurements in the pharyngoesophageal segment during swallows: unidirectional data were collected using a standard LRM catheter on 10 healthy subjects. The resulting recorded pressures were compared with circumferentially integrated data collected on the same subjects with a HRM system. Substantial differences were found for UES function with higher pressure amplitude and longer relaxation time recorded with the circumferential measurement system. Nevertheless, the performed circumferential measurements were the result of an integration on the sensor area: no information is available on the radial direction of the pressure stimuli yet.

Solid-state catheters are considered easier to set up and use but more fragile. Moreover, solid-state probes tend to be less flexible, thinner, and more expensive compared to water-perfused catheters [56].

HRM led to the development of the Chicago Classification for primary esophageal motility disorders [55], and it is now considered the gold standard in evaluating esophageal motor functions. The Chicago Classification enables the categorization of esophageal motility disorders by defining the standard parameters to be assessed during a high-resolution manometry (HRM) procedure, along with the specific values uniquely linked to various pathological conditions. Nevertheless, normative values have been defined using a solid-state HRM catheter with specific dimensional parameters, and many attempts have been made to verify if those values are still consistent with systems characterized by different features (i.e., diameter, number, and type of sensors). Xiang et al. [59] investigated whether the catheter diameter may be a device-dependent problem that influences the manometric results in a standard HRM procedure. In this study, HRM was performed on 9 asymptomatic volunteers and 18 patients affected by GERD. All the subjects were examined with two solid state catheters having different outer diameters (4.2 mm and 2.7 mm, respectively). The parameters defined by the Chicago Classification to make a diagnosis were extracted for both systems and compared. Xiang et al. showed that the 2.7 mm thick solid-state high-resolution manometry catheter provided results significantly different from the conventionally-used 4.2 mm thick catheter: the final diagnosis of 13 over 27 subjects was different when using two catheters that only differed for the outer diameter. This result strongly suggests the need to set up different and independent normative values for the solid-state catheters of different outer diameters.

Also, the choice of the technology employed to perform the HRM procedure influences the normative values measured and possibly the final diagnosis made by the clinician. Gehwolf et al. [60] compared LES pressure measurements acquired with a solid-state catheter and with a water-perfused one. The results obtained on 27 healthy volunteers showed that there was a significant difference between the LES pressure value evaluated with the two different technologies, with the solid-state manometry estimating a higher value of pressure.

Currently, solid-state technology is more frequently used in clinical practice even if the water-perfused system has not been abandoned. In 2020, Mariotto et al. [61] proposed a novel low-cost, water-perfused HRM system with a unique peristaltic pump and a helicoidal sensor distribution and demonstrated that the system was able to discriminate most of the motor disorders classified in the Chicago Classification.

Hee Man Kim [62] proposed a water-perfused catheter with the integration of an optical sensor on the tip for visual assistance in catheter introduction, thus preventing the coiling of the probe. Although the system was never adopted in clinical practice, it has been tested in-vivo, reaching a TRL 7 (estimated by the authors). The device presented by Hee Man Kim [62] still shows some limitations, such as: (i) lack of a lens cleaning system, (ii) air inflation, and (iii) low resolution of the images acquired. On the other side, the proposed device represents a promising improvement that could be helpful in clinics, especially for catheter placement in patients affected by hiatal hernia or other anatomical abnormalities.

The American Gastroenterological Association (AGA) developed official recommendations to assist physicians in the appropriate use of HRM in patient care, thus regulating when its use is appropriate and required [63].

To conclude, it is worth noticing that new technologies have evolved to complement different manometric diagnoses, such as the integration of impedance and pH sensors inside manometric catheters; further details are reported in Section 5.

As already mentioned, HRM is the most used procedure to assess the contractility behaviour of the esophageal lumen. Nevertheless, more recent techniques emerged in recent decades and are described in the next sections.

### 4.2. Functional Lumen Imaging Probe (FLIP) Planimetry

FLIP planimetry is a technique developed to assess simultaneously LES cross-sectional area (CSA) and pressure during contractility, thus evaluating sphincter distensibility and compliance.

The most used commercially developed FLIP system is EndoFlip^®^ (Endo Functional Luminal Imaging Probe system; Crospon Ltd., Galway, Ireland), shown in Figure 4b. It consists of a 100 mm-long polyurethane balloon mounted on a probe distal end. When filled with a special conductive solution, it reaches a maximum volume of 60 mL and a maximum diameter of 25 mm. The balloon is placed in correspondence to the LES and the nearby area, and it allows to perform 16 CSA measurements using an impedance planimetry technique; the balloon contains an array of 17 paired ring electrodes with a 5 mm pitch and excitation electrodes to obtain 16 impedance measurements. The probe also contains a solid-state pressure transducer to perform intra-balloon pressure measurements. The EndoFlip^®^ system provides a 3D real-time dynamic reconstruction of LES distension and the pressure reached during contraction.

Characterizing the mechanical properties of the LES is of great interest, as it largely affects the reflux barrier that should avoid refluxes events to occur. Excessive LES compliance allows greater volumes of gastric content to reflux into the esophagus, and it is indeed often associated with GERD disease. Kwiatek et al. [64] demonstrated that the commercially-available EndoFlip^®^ can discriminate healthy subjects from GERD patients by evaluating LES distensibility. The study was conducted on 20 control subjects and 20 GERD-affected volunteers during a routine esophagogastroduodenoscopy, and LES distensibility was studied with 10 to 40 mL of balloon volume filling. The results obtained were comparable with the values measured with previously used barostat-based devices, with a significative higher distensibility associated with the LES of GERD patients.

Acharya et al. [65] evaluated distal esophagus bio-mechanical activity using FLIP planimetry data extracted during sedated endoscopy on 85 volunteers, including 14 asymptomatic controls and a clinical cohort of 71 patients. In detail, two different metrics were assessed: (i) active work and (ii) work capacity, the first one being evaluated with moderate balloon volumes (≤ 40 mL, where contraction generates significant changes in CSA), and the second one with higher balloon volumes (≥60 mL, where contraction cannot significantly alter lumen CSA). Changes in luminal CSA and pressures were treated as displacement and force, respectively, to compute the energy spent during secondary peristalsis. Acharya et al. [65] demonstrated that data generated with a FLIP system were useful to characterize esophageal tract bio-mechanical activity highlighting statistically significant differences between healthy subjects and patients affected by diverse esophageal diseases.

EndoFlip^®^ has been commercially available since 2009, and, even if its usage had limited penetrance into clinical settings, there are several studies in the literature supporting its utility in general clinical practice.

Carslon et al. in [66] conducted a prospective multi-centrum study to demonstrate that the EndoFlip^®^ system provides a suitable and well-tolerated esophageal motility assessment at the time of endoscopy, meaning that the FLIP planimetry technique is able to anticipate the results investigated with HRM. In detail, Carlson et al. showed that the EndoFlip^®^ evaluation, performed on 40 patients (referred for endoscopy with a plan for future HRM), could detect abnormal esophageal motility at the endoscopic encounter (then confirmed with a standard HRM procedure). Additionally, normal motility on FLIP technique evaluation was predictive of a benign HRM.

Moreover, even if the FLIP planimetry technique was originally developed for the evaluation of LES bio-mechanical properties, Regan et al. [67] suggested that the EndoFlip^®^ system could also be used as a technique to study UES distensibility, and this could be of clinical relevance for better understanding inefficient bolus clearance in subjects with dysphagia.

### 4.3. Endoscopic Pressure Study Integrated System (EPSIS)

Another method to evaluate the bio-mechanical functions of LES is EPSIS, which consists of monitoring intra-gastric pressure (IGP) while insufflating the stomach during a gastroscopic procedure. A flat waveform pattern of IGP during stomach insufflation means that ${\mathrm{CO}}_{2}$ is continuously released during insufflation, and, thus, it is associated with an impaired LES. On the contrary, if the IGP waveform evaluated during EPSIS has an uphill pattern, it means that the LES can withstand the rising pressure applied during insufflation, and, therefore, it is associated with a physiological condition Figure 4c.

Shimamura et al. [68] conducted a retrospective analysis of patients with typical GERD symptoms and demonstrated that, by characterizing IGP waveforms with four different parameters (i.e., (i) Basal IGP, (ii) maximum IGP, (iii) pressure difference, and (iv) gradient of the waveform), abnormal presence of acid reflux could be assessed with high diagnostic accuracy.

Initially, an EPSIS system was developed as a catheter-based technique: an intragastric probe connected to an external pressure measurement system was inserted inside the working channel of an endoscope. In this regard, to reduce costs and simplify the procedure, the same authors of the previously mentioned work [68] also proposed a new, simplified system to perform EPSIS with no need for an additional probe [69]. In this work, the authors assessed the feasibility of an updated EPSIS system, which could be performed just by connecting a flush tube to the working channel of the endoscope and demonstrated that the performances of such a simplified system were comparable to the catheter-based one.

### 4.4. Others

In this sub-section, other methods to evaluate bio-mechanical properties of the esophagus, still at a research level, are reported.

Lu et al. [70] proposed a novel system to assess dynamic real-time monitoring of LES using a catheter-based acoustic device. The system consists of a micro-oscillator, located at one side of the LES, which actively emits sound waves at 16 kHz, and a miniature microphone located at the other side to capture the sound generated from the oscillator. In this way, the device could monitor the dynamics of the opening and closing of the LES. The system reached an estimated TRL of 6, being tested both in-vitro and in-vivo in a pilot canine model. The in-vitro test demonstrated a high correlation between the LES opening, detected by the acoustic system, and the LES simulator opening, monitored by a custom system. The in-vivo study likewise confirmed those promising results: the canine LES was forced to open and close by a transoral endoscope, which was monitored in real-time by a transpyloric endoscope inserted from the duodenum and positioned into the distal stomach. Frame-by-frame video analysis validated the interrelation between the sound strength and the LES opening and closing.

Santander et al. [71] suggested that another useful parameter to be evaluated for a better understanding of esophageal motor diseases was the overall muscle thickness, which, if abnormally increased, could lead to esophageal contractility dysfunctions. The authors indicated high frequency intraluminal ultrasound (HFIU) as the technique to employ for muscle thickness assessment, thus using a flexible ultrasound (US) catheter to characterize esophagus wall thickness.

**Figure 4 sensors-23-08858-f004:**
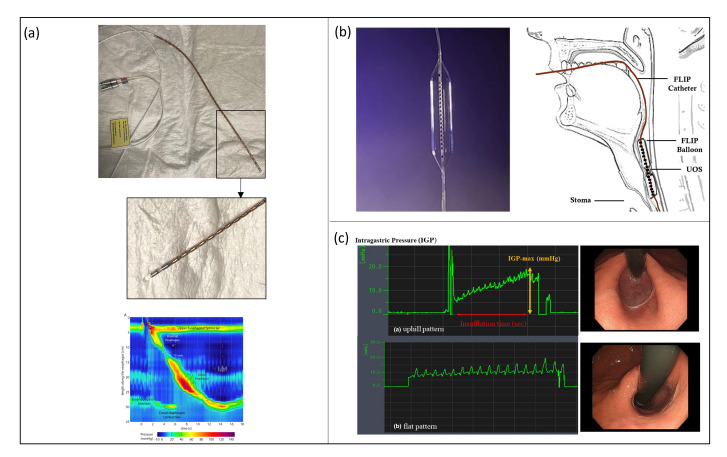
(**a**—**top**) HRM system consisting of a manometric solid state catheter (ManoScan™, Medtronic Inc., Minneapolis, MN, USA) (Courtesy of N.d.B.) and (**a**—**bottom**) a signal acquisition system and a GUI used by the clinician to perform the diagnosis based on a colormap; (**b**) EndoFlip^®^ (Endo Functional Luminal Imaging Probe system; Crospon Ltd., Galway, Ireland) [72]; (**c**) IGP waveforms of a physiological and pathological condition, respectively; study performed with an EPSIS procedure [69].

## 5. Endoluminal Technologies for Electro-Chemical Evaluation of Esophagus

Esophageal symptoms are common and difficult to be associated with a specific disease, since similar symptoms can arise from very different causes. Assuming that diseases with similar symptomatology and dissimilar etiology require different treatments, it is critical to follow a validated diagnostic process involving more than a single diagnostic step. For this reason, the technologies discussed above may not be sufficient to deeply understand esophageal health conditions. Other aspects to be analysed, in order to make assertions about esophagus health conditions, are the ones related to the electro-chemical properties of the esophageal lumen. In this regard, the following sub-sections investigate the endoluminal technologies available in literature for electro-chemical evaluations of the esophagus.

### 5.1. pH-Metry

A quantitative measure of esophageal exposure to acid can be performed by dynamically monitoring the pH level in different locations inside the esophageal lumen. In detail, the equipment used for conventional pH-metry involves the use of a catheter with two antimony sensors (20 mm away from each other) for pH evaluation and an external portable recorder. The catheter is placed trans-nasally inside the patient until the sensors are located, respectively, 30 and 50 mm above the LES (Figure 5(aI)). The probe is left in place for 24 h, and the recording device monitors and saves pH data during the entire procedure. The information extracted from a pH-metry procedure is the number of refluxate events and the acid exposure time (AET), defined as the time percentage of esophageal pH < 4 over the whole monitoring time. Those parameters allow the clinician to claim evidence of GERD. pH monitoring for 24 h can detect gastroesophageal reflux disease with a sensitivity and specificity of 87% and 97%, respectively [73]. Nevertheless, the main limit associated with catheter-based esophageal pH-metry relies on the fact that, during the monitored period, the patient changes their daily habits due to discomfort caused by the catheter presence: they may reduce food intake and behave differently than usual, thus influencing the diagnosis. To overcome this problem, wireless catheter-less systems for pH monitoring have also been developed. Conventionally, 24 h wireless ph-metry is performed with a miniaturized capsule—provided with an antimony pH sensor—placed with a suction anchoring system 50 mm above the LES. The measured data are transmitted via telemetry to the external recorder. Capsule detachment occurs spontaneously after some days. The gold standard capsules available in clinical practice and commercialized by Medtronic Inc. are the BRAVO^TM^ capsules (Figure 5(aII)).

In [74], Azzam et al. carried out a comparative study between catheter-based and capsule-based systems: 25 volunteers with suspected GERD were tested with both technologies and then subjected to a clinical questionnaire. The authors concluded that, although better tolerability is associated with the wireless system, such a solution is more expensive, and there is no evidence of higher sensitivity to GERD.

Nevertheless, it is worth underlying that the wireless system can be adopted for a longer monitoring time (typically 48 h) than the wired one. In this regard, Domingues et al. [75] investigated the impact of a prolonged monitoring time in the diagnosis of GERD with particular concern on day-by-day variability in esophageal acid exposure and patient tolerability of the capsule implantation. The study, conducted on 100 patients, revealed a significant daily variability of symptoms, yielding an increase in GERD diagnosis of 43.4% in patients that would be otherwise missed in a 24 h study. On the other hand, nearly 15% of subjects who underwent wireless pH capsule placement complained of symptoms ranging from foreign object sensation to chest pain.

Regardless of the technology adopted, the main limitations of pH-metry in the diagnosis of GERD relied on the ability to detect only acid refluxates, not considering the impact of non-acid reflux events. To overcome this limit, impedance measurements have been integrated into a next generation of devices illustrated in the next section (Section 5.2).

### 5.2. Multichannel Intraluminal Impedance and pH (MII-pH) Monitoring

To allow for the detection of both acid and non-acid gastroesophageal reflux episodes, the MII-pH technique combines pH and impedance measurements to monitor bolus transit, discriminating between the antegrade and retrograde direction of both acid and non-acid materials. The working principle is the following: the endoluminal electrical conductivity of the esophagus increases during the passage of a bolus (i.e., food, saliva, or gastrointestinal secretions), whereas it decreases when the organ is collapsed (or filled with air). Multiple impedance channels, located at different points along the esophagus lumen, detect the electrical conductivity in the surrounding environment, thus localizing and monitoring bolus movements inside the organ. Simultaneously, the pH channels characterize the chemical nature of the bolus detected. The MII-pH monitoring procedure is now considered the gold standard in the diagnosis of GERD.

In detail, a standard catheter for MII-pH monitoring procedure includes six impedance segments placed 30, 50, 70, 90, 150, and 170 mm above the lower esophageal sphincter (LES), together with a pH electrode at 50 mm (Figure 5b).

Usually, before an MII-pH study, the patient undergoes a manometry procedure to exclude esophageal transit abnormalities and to accurately localize LES, which is the reference point for the correct placement of MII-pH catheter. This introduces a considerable limit both in terms of costs and patient tolerability, since it implies two procedures to be performed on the same subject. In this regard, Amine Hila et al. [76] demonstrated that complete bolus transit and swallow evaluation were assessable by assessing 10 saline swallows in the recumbent position at the beginning of the MII-pH study, obtaining a sensitivity of 94% and a specificity of 93% for detection of esophageal transit abnormalities. The authors concluded that there may be no need of resorting to additional manometric studies before performing an MII-pH procedure to determine if the patient suffers from some kind of esophageal transit abnormality.

On the other hand, as mentioned before, manometry is still required for the accurate localization of LES for MII-pH catheter placement, bringing additional time, cost, and discomfort associated with dual nasal intubation. An alternative method to assess LES location is the airflow sphincter locator (AFSL) system, manufactured and commercialized by Sandhill Scientific Inc. (Highlands Ranch, CO, USA): an external perfusion pump is connected to a MII-pH catheter and generates a constant pneumatic pressure through a distal port. The catheter is placed trans-nasally until the tip reaches the stomach, and a pressure offset is performed. Pressure is monitored through an external recorder while the catheter is pulled back 10 mm at a time: the position at which the pressure reaches a positive value is associated with the LES location. Although the AFSL system reached the market, it was never widely adopted in clinical practice due to its low accuracy. Indeed, Chen et al. [77] claimed that the AFLS system is not an acceptable alternative to HRM in LES localization: the study was conducted on 50 subjects, and the LES position was assessed both with AFSL and HRM. The authors demonstrated that the AFSL system placed the LES outside of the ±30 mm range was considered clinically-acceptable in 32% of the patients.

As for pH-metry procedures, wireless systems are being investigated also for MII-pH monitoring. To do that, impedance measurements must rely on fewer channels to be integrated on a shorter device that can be somehow anchored to the esophagus lumen. R. Heard et al. [78] aimed to verify if the direction of bolus movement could be reliably determined using only two impedance channels, and, if so, which two channels were the most suitable ones. In total, 20 patients underwent MII-pH monitoring, and the results were evaluated considering only two over the six impedance channels and one over the two pH sensors. In detail, three different combinations of impedance channels were investigated: (i) 30 mm and 50 mm, (ii) 30 mm and 70 mm, and (iii) 50 mm and 70 mm above the LES. The authors concluded that the most reliable configuration was the one with the impedance channels located, respectively, 30 mm and 70 mm above the LES. Nevertheless, the use of a shorter, wireless MII-pH catheter came with considerable disadvantages, such as the inability to evaluate the proximal migration of refluxes and difficulty to differentiate between swallows and reflux episodes (due to the lower number of impedance sensors).

On the same topic, Hung Cao et al. [79] developed an implantable wireless and battery-free capsule for pH and impedance monitoring. The capsule was encapsulated in a biocompatible polymer and anchored on the esophageal wall by endoscopic procedures. An external portable reader, embedded in a wearable belt, was also provided. Both powering and signal transmission were achieved through wireless electromagnetic coupling between two coil antennas located, respectively, on the implantable device and the external portable reader. To evaluate the system performances, the device was tested both in vitro using a mannequin of a human esophagus and in-vivo using live pigs, reaching an estimated TRL of 7.

As already mentioned, catheters provided with multiple kinds of sensors (e.g., pressure sensors together with pH and impedance sensors) are emerging in clinical practice to combine different analysis in a single procedure. Table 1 presents a compilation of widely utilized commercial devices, showcasing their key features for easy reference.

### 5.3. Bilimetry

Another parameter worth investigating for the assessment of upper GI tract health condition is the presence and concentration of bile refluxes. It has indeed been recently demonstrated that the presence of bile in an acidic esophageal environment is associated with more severe heartburn [80].

Bilitec™ 2000 (Cecchi s.r.l., Florence, Italy) is an endoluminal probe provided with a fiberoptic spectrophotometer embedded inside. The working principle is based on the property of bilirubin absorbing light at a specific wavelength. In detail, two light-emitting diodes are integrated inside the tip of the probe: one blue (emitting at 470 nm, close to the peak bilirubin absorbance of 450 nm), and a reference green diode (565 nm). The light is transmitted in a 2 mm gap in the head of the probe and is reflected by a white polyvinyl chloride cap. By measuring the difference in absorption between the two emitted wavelengths, the concentration of bilirubin in the refluxate inside the gap can be determined as the blue light will be absorbed proportionally to the concentration of bilirubin (Beer–Lambert Law, shown in Equation (3)) (Figure 5c).
A = ε·c·l,(3)
where A is absorbance, ε is molar absorption coefficient [$M^{-1}{\mathrm{cm}}^{-1}$], c is molar concentration (M), and l is the optical length (cm).

As mentioned in [81], possible limitations of the technology may be the following: (i) non-clearance of the probe’s sensor region, in which food particles and viscous material may remain longer than in the surrounding mucosa [82], (ii) necessity to undergo a white diet during the monitoring time to do not affect absorbance, which may introduce a change in the normal gastrointestinal behaviour aimed to be characterized, and (iii) decreased sensitivity when applied in-vivo compared to in-vitro validation studies.

Innovative, yet low, TRL alternatives are available in scientific literature. In [81], Dhiren Nehra proposed a model for developing a biosensor able to detect the presence of bile acid. The idea is to use Molecular Imprinting Technology (MIT), i.e., a technique based on creating artificial recognition sites in polymeric matrices that are complementary to the target (i.e., bile acid) in their sizes, shapes, and spatial arrangements of the functional groups [83]. The system would be integrable onto existing pH endoluminal probes, and contrary to spectrophotometry, it would have high specificity and sensitivity. Although biosensor integration on existing devices is a promising field for endoluminal investigation, the practical application of this approach is still limited due to the complexity of the biosensor development process.

### 5.4. Mucosal Impedance Test (MIT)

Most of the diagnostic tools to assess GERD (i.e., pH-metry, MII-pH procedures, and HRM) are constrained by the limited time of observation and cannot provide a measure of chronicity of reflux and long-term effects on esophageal mucosa. Therefore, new methods to assess esophageal epithelial integrity—without resorting to invasive biopsies—are being investigated and have provided prolific literature in recent decades [84].

In detail, it is a common opinion within the medical community that direct measurement of esophageal epithelial integrity employing mucosal impedance (MI) has the potential to reliably detect GERD and EoE. Dilation of intercellular spaces (DIS) between esophageal epithelial cells is a common condition in patients affected by GERD and EoE, and the degree of DIS is shown to inversely correlate with MI measurements. Thus, MI can be used as a marker of histological changes in patients, and it can be correlated to a GERD or EoE condition.

Many studies attempted to demonstrate the efficacy of MIT in clinical practice [85,86,87,88,89]. Yuksel et al. [85] designed a new catheter containing an array of two impedance sensors with a 2 mm pitch to be inserted throughout the operative channel of a gastroscope and be placed in direct contact with the esophageal endoluminal wall to perform MIT at the time of upper endoscopy. A prospective study was conducted on a heterogeneous cohort of patients with (i) erosive GERD (n = 19), (ii) non-erosive GERD (n = 23), and (iii) control subjects (n = 27), resulting in significantly lower values of MIT at the sites of eroded mucosa. Many other studies suggest that performing MIT at the time of endoscopy using ad hoc re-designed impedance catheters may lead to significative evidence of GERD and may help to discriminate between erosive and non-erosive diseases [87,88,89].

Similar conclusions were achieved by Patel et al. in [89] and by Jeffrey A. Alexander et al. in [90] using balloon MI catheters. This family of devices allows the clinician to obtain MI measurements on a segment of the esophagus through an array of impedance sensors integrated inside a biocompatible balloon to reach perfect compliance with the esophagus inner walls. Patel and colleagues used a balloon probe developed by Diversatek Healthcare Inc. (Milwaukee, WI, USA) (Figure 5d), with 2 columns of 9 sensors with 10 mm pitch separated by 180 degrees intervals for a total amount of 18 sensors and a sensitive length of 100 mm. Jeffrey A. Alexander et al. [90] used an MI custom balloon assembly with a balloon of 110 mm in length provided with 2 axial arrays of 10, for a total amount of 20 sensitive elements. Both the aforementioned studies found a positive correlation between lower values of MIT and pathological impaired condition of esophageal mucosa. Finally, in [91], Gaurav et al. proposed a wireless version of an MIT device, developing a gelatine-based ingestible capsule to monitor epithelial barriers via electro-chemical impedance measurements. The device, still at a very low TRL, represents a promising alternative towards less invasive procedures.

## 6. Conclusions

The past few decades have witnessed substantial evolution of heterogeneous and, sometimes, multimodal techniques to evaluate the esophageal tract health condition.

The continuous clinical translation of novel systems to enable in-vivo characterization of the esophageal tract paved the way to new opportunities for improved detection of upper GI diseases, potentially leading to important turning points in diagnostic and therapeutic algorithms.

Evaluation of the esophageal tract health condition requires a multidisciplinary and integrated effort of many different techniques and procedures including: (i) imaging techniques (i.e., WLE, NBI, MSI, AFI, SERS, and endomicroscopy), (ii) functional tests (i.e., HRM, FLIP planimetry, and EPSIS), and (iii) procedures aimed at characterizing electro-chemical properties of the lumen (i.e., pH-metry, MII-pH monitoring, bilimetry, and MIT). This varied and constantly evolving landscape of modern diagnosis presents numerous opportunities on one side but, on the other side, may lead to difficulties in standardizing diagnostic algorithms and procedures to be followed by clinicians.

Therefore, in this review, with any claim of suggesting the most appropriate workflow to be followed in the diagnosis of the esophageal tract-related diseases, authors intended to provide a comprehensive analysis of the various diagnostic procedures, endoluminal technologies, sensing principles and commercial-/research-oriented devices. In addition, this work is intended to support the scientific community’s understanding on the heterogeneous pathological processes of esophageal diseases to highlight the importance of adopting multidisciplinary and integrated approaches in addressing the healthcare scenarios.

In particular, due to advanced technologies, superior accuracies, and heightened diagnostic capabilities, imaging procedures are considered the primary diagnostic mean of esophageal diseases. Nevertheless, such procedures are the most invasive and expensive ones. Indeed, after an initial screening phase involving less invasive and cost-effective techniques, imaging is foreseen to be employed only for patients who have already been confirmed as suspects of having a disease. In this regard, imaging procedures are trying to evolve towards less invasiveness. The realization of such a scenario will rely on improvements in the accuracies of functional analyses, with HRM being the most promising technique, as well as on advancements in electro-chemical evaluations. Consequently, these non-imaging techniques may be effectively utilized for the initial screening phase, as well as for subsequent follow-up procedures. 

Moreover, it is worth mentioning that the presented technologies rely on direct data processing only, without applying AI-based algorithms. In this regard, we expect that ongoing and future developments in AI and big data analysis will significantly impact these diagnostic techniques, with the potential to enhance clinical capabilities with improved prevention, prediction, and personalized therapies.

## Figures and Tables

**Figure 1 sensors-23-08858-f001:**
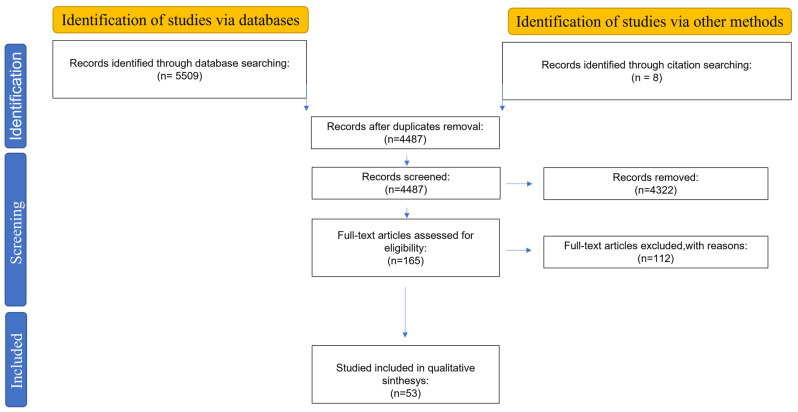
Outline of the followed literature review process with PRISMA diagram.

**Figure 2 sensors-23-08858-f002:**
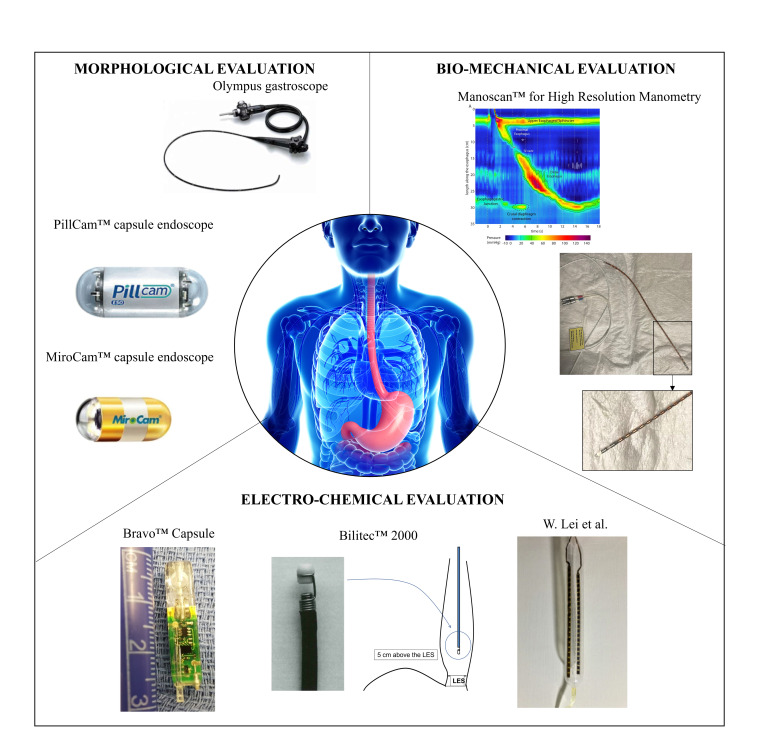
Examples for the investigated categories for morphological, bio-mechanical, and electro-chemical evaluation, which will be further detailed in the next sections. Olympus gastroscope (GIF XQ240, Olympus Corporation, Tokyo, Japan) was selected as a representative example of the conventional white light endoscopy section (Courtesy of Olympus Corp.); PillCam^®^ capsule endoscope by (Medtronic Inc., Minneapolis, MN, USA) is the most clinically-used device in the field of passive wireless endoscopes [14]; MiroCam^®^ capsule endoscope (IntroMedic Co., Ltd., Seoul, Republic of Korea), as an example of magnetically actuated capsules [14]; Bravo™ (Medtronic Inc., Minneapolis, MN, USA) for wireless pH monitoring [15]; Bilitec™ 2000 for bile detection (Cecchi s.r.l., Florence, Italy); catheter for mucosal impedance test by W. Lei et al. [16]; and ManoScan™ for High Resolution Manometry (Medtronic Inc., Minneapolis, MN, USA) (Courtesy of N.d.B.).

**Figure 5 sensors-23-08858-f005:**
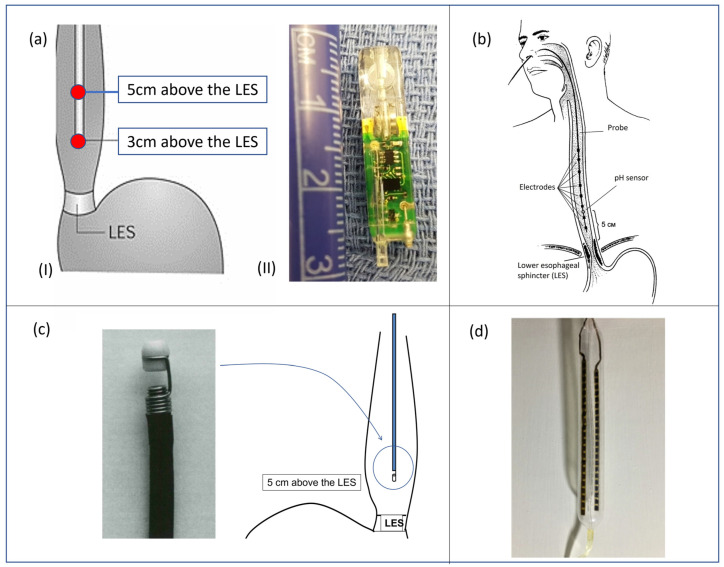
(**aI**) pH-metry catheter correctly positioned inside the esophagus lumen and (**aII**) BRAVO^TM^ capsule by Medtronic Inc. (Minneapolis, MN, USA); (**b**) standard catheter for MII-pH, correctly placed for a gastroesophageal procedure; (**c**) Bilitec™ 2000 system, produced by Cecchi s.r.l. (Florence, Italy) for bile detection; (**d**) balloon probe developed by Diversatek Healthcare Inc. (Milwaukee, WI, USA) for MIT procedure [16].

**Table 1 sensors-23-08858-t001:** Examples of the most used commercial devices for HRM, HRM combined with impedance measurements, and MII-pH metry; N/A: not applicable, N/D: not declared.

Device		ManoScan™ for HRM	ManoScan™ for HRIM	DIGITRAPPER™ pH-Z	Unisensor AG for HRM	G-HRIM
Manufacturer		Medtronic Inc.	Medtronic Inc.	Medtronic Inc.	Laborie Medical Technologies Inc.	Sandhill Scientific Inc.
Diameter (mm)		4.2	4.2	1.57	4	4
Pressure	N. of pressure channels	Up to 36	Up to 36	-	Up to 36	Up to 32
	Gap (mm)	10	10	N/A	10	10
	Type of sensors	Solid state	Solid state	N/A	Solid state	Solid state
	Type of measurement	Circumferential	Circumferential	N/A	Circumferential	Circumferential
	Resolution (mmHg)	0.01	0.01	N/A	N/D	N/D
	Update rate (Hz)	100	100	N/A	N/D	N/D
Impedance	N. of impedance channels	-	18	8	-	16
	Distance from LES	N/A	N/D	−3, −1, 1, 3, 5, 9, 11, 13	N/A	N/D
pH	N. of pH channels	-	-	2	-	-
	Distance from LES (mm)	N/A	N/A	−150, 0	N/A	N/A

## Data Availability

Data sharing not applicable.

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
