# Peer review of "Endoluminal Procedures and Devices for Esophageal Tract Investigation: A Critical Review"

_sensors, 2023, doi:10.3390/s23218858_

Round 1

Reviewer 1 Report

The article is interesting and valuable. The work contains a description of important devices used in the diagnosis of the gastrointestinal tract. The article is important to collegoues working in the field. The work is intended for people involved in research on the human digestive tract. The work is thematically coherent. No errors in reasoning.Due to the lack of a mathematical description and the lack of proposals for new solutions in diagnostic equipment, the work is not intellectually stimulating The aesthetics of the Figures is proper. The conclusions are correct. The references are modern.

Major:

You sholud add at least 5 modern references to introduction section.

The serious lack of the article is lack of mathematical formulas described the physical phenomena in diagnostic devices.

The article does not describe the electronics used in the relevant diagnostic devices

Minor:

You should put some text below Figure 2

Conclusions:

The work is an example of a descriptive article. There is no description of own research specifying the disadvantages and disadvantages of the devices used in the diagnosis of the digestive system.

The article checked by a professional linguist is much more accessible to the Reader.

Reviewer 2 Report

I found the Review article authored by Spreafico et al. to be highly engaging. The manuscript demonstrates excellent writing and a clear organizational structure. The inclusion of relevant figures and schemas enhances the reader's understanding of the content. I would like to offer a minor suggestion to the authors, which is to delve into a discussion of how the technology they present not only influences the diagnostic process but also contributes to a deeper understanding of the pathological processes involved in upper gastrointestinal diseases.

Minor orthographical and grammatical errors have been found throughout the manuscript.

Reviewer 3 Report

It is a systematic review article on the UGI search device, including morphological and functional aspects. It is interesting since this type of article is scary. However, in my opinion, there are some concerns about this article. 1. The authors should clearly define the meaning of UGI(e.g., the oral side of the Treitz ligament) in this review article. 2. The article did not describe enough on the small intestine, including capsule endoscopy, as much as the esophagus issue. 3. The author could add the AI aspect of section this issue. 

Minor English editing isrequired. 
